# Perceived usefulness of trauma audit filters in urban India: a mixed-methods multicentre Delphi study comparing filters from the WHO and low and middle-income countries

Johanna Berg [1,2] Helle Molsted Alvesson [1] Nobhojit Roy [1,3] Ulf Ekelund,[4] Lovenish Bains [5,6] Shamita Chatterjee,[7] Prosanta Kumar Bhattacharjee,[8] Siddarth David,[1,9] Swati Gupta,[10] Jyoti Kamble,[9,11] Monty Khajanchi,[6,12] Pawanindra Lal,[5] Vikas Malhotra,[13] Ravi Meher,[13] Anurag Mishra,[5] Lakshmeswar Nagaraj Mohan [14] Max Petzold,[15] Ritu Saxena,[16] Prabhat Shrivastava,[17] Rajdeep Singh,[5] Kapil Dev Soni,[18] Sumit Sural,[19] Martin Gerdin Wärnberg [1,20]

For numbered affiliations see end of article.

**Correspondence to**
Johanna Berg;
johanna.berg@ki.se

## ABSTRACT

**Objective** To compare experts' perceived usefulness of audit filters from Ghana, Cameroon, WHO and those locally developed; generate context-appropriate audit filters for trauma care in selected hospitals in urban India; and explore characteristics of audit filters that correlate to perceived usefulness.

**Design** A mixed-methods approach using a multicentre online Delphi technique.

**Setting** Two large tertiary hospitals in urban India.

**Methods** Filters were rated on a scale from 1 to 10 in terms of perceived usefulness, with the option to add new filters and comments. The filters were categorised into three groups depending on their origin: low and middle-income countries (LMIC), WHO and New (locally developed), and their scores compared. Significance was determined using Kruskal-Wallis test followed by Wilcoxon rank-sum test. We performed a content analysis of the comments.

**Results** 26 predefined and 15 new filter suggestions were evaluated. The filters had high usefulness scores (mean overall score 9.01 of 10), with the LMIC filters having significantly higher scores compared with those from WHO and those newly added. Three themes were identified in the content analysis relating to *medical relevance*, *feasibility* and *specificity*.

**Conclusions** Audit filters from other LMICs were deemed highly useful in the urban India context. This may indicate that the transferability of defined trauma audit filters between similar contexts is high and that these can provide a starting point when implemented as part of trauma quality improvement programmes in low-resource settings.

## STRENGTHS AND LIMITATIONS OF THIS STUDY

⇒ This study is the first to evaluate the usefulness of trauma audit filters from different sources in a low-resource context.
⇒ We used two separate study sites to perform parallel independent online Delphi surveys to compare the usefulness of audit filters for trauma care.
⇒ Both sites scored and selected filters with the intent to implement them.
⇒ The content analysis gave a deeper understanding of the characteristics of the audit filters.
⇒ Due to the nature of the Delphi methodology, the statistical analysis is based on a convenience sample.

4.4 million deaths globally every year.[1] With nearly 90% of these occurring in low and middle-income countries (LMIC), it has been estimated that over 2 million lives each year could be saved if fatality rates were reduced to the levels of high-income countries (HIC).[2] Strengthening trauma care is essential to the global health agenda and there is evidence that implementing trauma quality improvement programmes decreases mortality and improves trauma care in both HIC and LMIC.[3–5] These programmes are diverse and involve different approaches to improving care, one of which is audit filters.

Audit filters, in some cases also referred to as quality indicators, are predefined statements that define deviations from standard care. For example, an audit filter could state, 'Patient with Glasgow Coma Scale score <8 should have an endotracheal tube or surgical

## INTRODUCTION

Trauma, defined as physical injury and the body's associated response, accounts for

**BMJ**

airway performed before leaving resuscitation area.' A case violating this filter would indicate suboptimal care and the case flagged for review. Even though there is evidence that trauma quality improvement programs improves outcomes, high-quality evidence on the effects of trauma audit filters is lacking.[6] The use of audit filters is one technique promoted in the WHO guidelines for trauma quality improvement programmes.[7] Their implementation requires more extensive resources, compared with other proposed interventions, since they are data driven.[7]

Audit filters need to be adapted to the context in which they are to be applied.[7 8] To generate context-relevant audit filters, previous studies have used the Delphi technique to identify appropriate filters from a group of experts.[9 10] There is limited experience evaluating the filters proposed by the WHO and their perceived usefulness in a low-resource setting.

India accounts for 20%–25% of trauma mortality globally.[1] In-hospital mortality remains high, with one study reporting 30-day mortality of 21%, which is twice as high compared with data from registries in HIC.[11] More than 50% of these deaths have been reported to be preventable.[12] Identifying evidence-based strategies to improve in-hospital trauma care in India is vital.

With this background, this study aims to compare experts' perceived usefulness of audit filters from Ghana, Cameroon, WHO and those locally developed; generate context-appropriate audit filters for trauma care in selected hospitals in urban India; and explore characteristics of audit filters that correlate to perceived usefulness.

## MATERIALS AND METHODS
### Design
This study was nested in the Trauma Audit Filter Trial (TAFT) (ClinicalTrials.gov, ID NCT03235388, preresults), which investigates the effect of audit filter implementation on mortality and morbidity in adult trauma patients in urban India.

We conducted a mixed-methods study based on a Delphi technique to facilitate consensus within a panel of experts.[13] The Delphi technique is based on structured, iterative, anonymous surveys where participants can rate and comment on statements. Between each iteration, researchers may report feedback to the panellists to facilitate the discussion. Definition of when consensus is reached in a Delphi is predefined but the exact methods used to define this vary between studies.

The Delphi technique has been used in multiple areas of healthcare research, including selecting trauma audit filters.[9 10] It is considered particularly useful when investigating complex multidisciplinary problems in areas with limited previous research where considerable uncertainties still exist.

This mixed-methods design allowed us to gather quantitative data from the scoring of the audit filters to answer the question of whether perceived usefulness differed based on the original source as well as select the highest scoring filters for implementation.[14] The qualitative data, the written comments deliberated during the Delphi rounds, were used to explore characteristics of the audit filters and their correlation to the usefulness scores.

We performed two independent Delphi surveys at two different hospitals to allow for the audit filters to be selected and modified based on local priorities and capabilities at each hospital. This design also allowed us to compare scores between two independent groups of experts and see if results were reproducible across the two study sites.

### Setting
We conducted this study at two public teaching hospitals of Kolkata and Delhi, both with populations of more than 15 million: Seth Sukhlal Karnani Memorial Hospital in Kolkata and Maulana Azad Medical College with associated Lok Nayak Hospital in Delhi. Both are public tertiary teaching hospitals that serve as referral hospitals. The study was conducted online between December 2018 and March 2019.

### Identification of audit filters
We included previously published trauma audit filters intended for use in LMICs. These filters included those from the WHO guidelines,[7] as well as filters in one study from Cameroon[9] and one from Ghana.[10] In total, 67 audit filters were identified with 47 from Cameroon[9] and Ghana[10] and 20 from WHO guidelines.[7] We did not deem it feasible to score and comment on all 67 filters at each site and therefore the India-based TAFT core research team conducted an internal online Delphi survey to remove duplicate filters, prioritise filters potentially appropriate for the urban setting in India and evaluate the online tool used for this study. The core research team consisted of five participants, four surgeons and one critical care/anaesthesia physician. All participants were at the professor or associate professor level and had extensive experience of clinical trauma work and research in urban India. The online Delphi for the core research team was performed in November 2018. The survey consisted of three rounds and response rates were 100% at all rounds. This resulted in 26 selected filters: 5 from WHO and 21 from LMIC sources (online supplemental table 1).

### Participants
We purposefully sampled physicians, surgeons, nurses and administrators with >5 years of professional experience and involvement in trauma care at the respective hospitals. This was facilitated through the ongoing TAFT project, where local principal investigators with intimate knowledge of the sites could identify experts for inclusion. We chose this approach since considerable knowledge about the local conditions and organisation, in addition to work experience, would be necessary to evaluate the usefulness of the filters at each site. Because we also allowed the filters to be modified and new filters to

be suggested, we deemed 10–15 participants feasible to reach a consensus. Before executing the online surveys, a 2-day meeting was arranged at the participating hospitals discussing trauma quality improvement programmes and introducing the concept of audit filters to the participants.

### The online surveys, consensus and stability criteria

We used the online survey software LimeSurvey to create an anonymous survey for each site.[15] Potential participants were invited to a website with information about the project, where they could register and then take part in the survey.

The audit filters were presented to the participants without revealing their sources. Rating of the perceived usefulness, defined as usefulness from the perspective of experts, was done on an ordinal Likert scale with grades from 1 to 10. A score of 1 represented not useful and 10 very useful. Open comments could be entered for each filter, and new filters could be proposed. Modifications to the filters could also be suggested. Comments were reviewed and summarised by members of the core research team after completion of each round; these were presented along with the median score of the filter in subsequent rounds.

Filters that received a median score of 7 or higher and newly added filters were included in subsequent rounds. Filters with a score of <7 were immediately rejected. A filter was deemed complete when it was either rejected or reached consensus for the selection, defined as having a median score of 9 or higher over two rounds, or when stable, defined as no significant changes in responses between rounds. A stable filter with a score of <9 was rejected. Stability was determined using the Wilcoxon rank-sum test, comparing scores from the previous and current rounds for each filter. A significance level of 0.05 was used. Filters modified based on comments from the panellists were included in subsequent rounds even if they had previously reached a consensus. The iterations continued until all remaining filters reached either consensus or stability. Filters that had a median score of 9 and above were selected for implementation at each site. Reminders were sent via email up to three times if no reply had been recorded.

### Statistical analysis of the results

We analysed the sites independently. To compare the perceived usefulness between different sources of audit filters, the filters were categorised into three groups based on their origin (LMIC, WHO and New). The filters defined in Cameroon and Ghana were combined as the group LMIC, the WHO guidelines as WHO and new filters proposed by the panellists at the sites as New. Filters that were modified were still included in the group of their origin.

The score from all participants in the final round for each filter was used for analysis. For example, if a filter was rejected in round 2, the scores from all participants in that round were used for analysis. If a filter was selected in round 3, scores from that round and filter were used. We used the Kruskal-Wallis test to compare usefulness scores across the three groups. If this resulted in a significant difference (p<0.05), post hoc testing was conducted using the Wilcoxon rank-sum test to identify differences between the individual groups using a significance level of p<0.05. We used the Bonferroni method to adjust p values for multiple tests. We used R for all analyses.[16]

### Content analysis

We explored the characteristics of the audit filters that were discussed using inductive content analysis.[17] We chose this approach as there are limited studies that have described characteristics of audit filters. The team of authors represent a multidisciplinary group of clinical, social science and epidemiological expertise. Several are physicians and surgeons at participating hospitals.

We collated the comments from both sites. For the initial analysis, the comments were organised so that only the comments and not the associated filters were visible. We did this to detach the comments from the filters and focus on the general problems or characteristics discussed. First, the entire collection of comments was read several times. Then meaning units were identified and abstracted to codes. All codes were sorted into categories that were abstracted into themes. After this, we assessed how the themes and categories compared with usefulness scores and the three source groups; WHO, LMIC and New. We did this by defining groups of low-scoring and high-scoring filters, based on their final filter scores. We then selected a theme or high-level category and assessed which filters and filter groups the theme was associated with. We used frequency analysis to see if the theme may be dominant in any of the specified groups. If this was the case, we returned to the comments associated with this theme and group to understand the specific discussion, from this perspective, better. We used this frequency analysis to help find patterns in the comments that would have been difficult to detect. We developed an audit trail as the coding was discussed between authors JB, HMA and MGW during the process of analysis. We used the NVivo software for analysis.[18]

### Patient and public involvement

We did not involve patients or the public in this study.

## RESULTS
### Participants

A total of 27 individuals were invited; of these, 25 agreed to participate. Participation remained high at both sites throughout rounds (mean participation/round 84%). The distribution of the professions differed across sites,

**Table 1** Characteristics of Delphi panellists, by site

| Characteristic | Site 1, n=10* | Site 2, n=15* |
|---|---|---|
| Profession | | |
| Administrator | 1 (10) | 0 (0) |
| Nurse | 2 (20) | 0 (0) |
| Physician | 1 (10) | 5 (33) |
| Surgeon | 6 (60) | 10 (67) |
| Department | | |
| Accident and emergency | 1 (10) | 2 (13) |
| Anaesthesia | 1 (10) | 2 (13) |
| Burns and plastic surgery | 0 (0) | 1 (6.7) |
| Critical care | 1 (10) | 0 (0) |
| Hospital administration | 1 (10) | 0 (0) |
| Orthopaedic surgery | 1 (10) | 2 (13) |
| Radiology | 0 (0) | 1 (6.7) |
| Surgery | 5 (50) | 7 (47) |
| Female/male ratio | | |
| Female | 4 (40) | 3 (20) |
| Male | 6 (60) | 12 (80) |

*n (%).

with one site being represented by surgeons and physicians only (table 1).

## Surveys

The online surveys were performed from December 2018 to March 2019. A total of four rounds were done at each site. The response rates remained stable with high participation rates for each round (site 1: 80%–100%, site 2: 93%).

## Perceived usefulness

The LMIC filters had the highest mean usefulness score at both sites (*site 1: 9.132, site 2: 9.568 of 10*). Comparing all three source groups, WHO, LMIC and New, we found significant differences in usefulness scores at both sites (*Kruskal-Wallis test: site 1 p<0.001, site 2 p<0.001*). Comparing pairwise between groups at each site, we found significant differences in usefulness scores between all groups, except between New and WHO filters at site 1 (*p=1*) (figure 1). Filters that were modified during rounds had no significant difference in scores compared with those that were not modified (*site 1: p=0.322, site 2: p=0.067*) (table 2).

## Selection of audit filters

In total, 30 filters for site 1 and 37 filters for site 2 were processed, resulting in a selection of 24 and 32 filters, respectively (figure 2).

## Characteristics of audit filters in relation to their usefulness

During the content analysis, three major themes relating to an audit filter's feasibility, medical relevance and specificity evolved. Creating audit filters appears to be a balance between medical relevance and feasibility as well as generality and specificity. Logistic limitations may force providers to modify standards of care to make something feasible. For example, if there is limited availability to an operation theatre (OT), someone needs to prioritise which patient should get access first. Defining this in audit filters is a balance between medical relevance (the most injured patient goes to OT first) and feasibility (limited to one OT, it might not be feasible to state that non-life-threatening injuries should have surgery within 6 hours). The filters need to be general enough to be

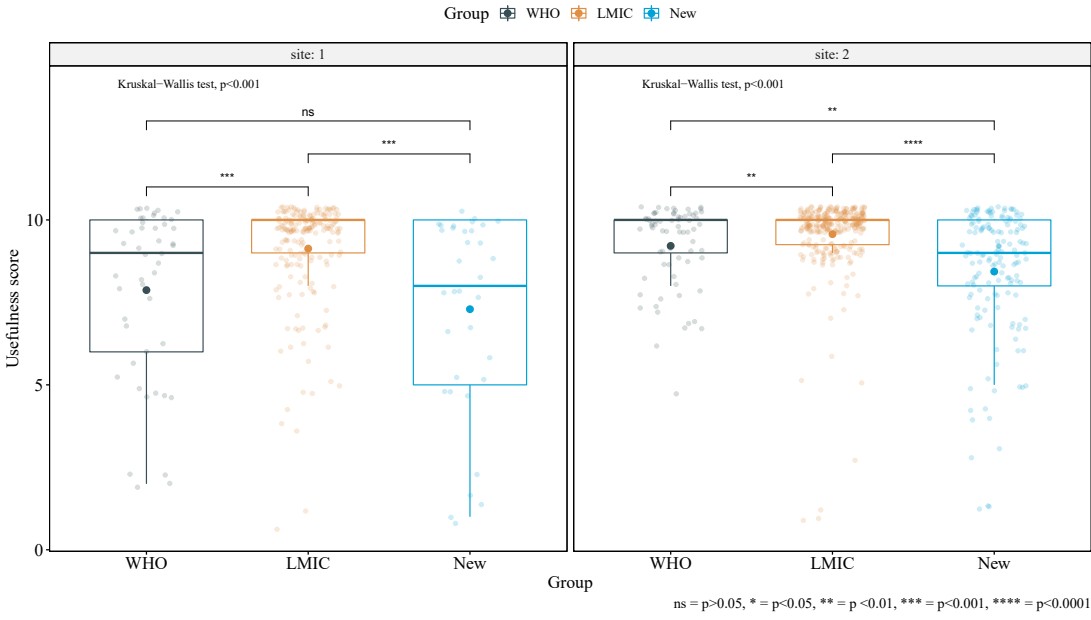

**Figure 1** Usefulness scores and level of significance for trauma audit filters by source group. LMIC, low and middle-income country.

**Table 2** All processed audit filters, source group and final scores at both sites (rejected filters are italic)

| Group* | ID | Audit filter | Score† | Site | Modified‡ |
|---|---|---|---|---|---|
| WHO | 3.1 | Hourly GCS in the emergency department of trauma patients with a diagnosis of skull fracture, intracranial injury or spinal cord injury. | 10.0 | 1 | Yes |
| | 3.2 | Sequential (every 30 min) GCS of trauma patients with a diagnosis of skull fracture, intracranial injury or spinal cord injury. | 10.0 | 2 | Yes |
| | 5 | Documentation of history and physical examination by a doctor. | 10.0 | 1 | No |
| | | | 10.0 | 2 | No |
| | *6.1* | *Head CT scan done within 2 hours of arrival at hospital for a non-transferred patient with Glasgow Coma Scale score <13 and systolic blood pressure >90.* | *6.5* | *1* | *No* |
| | 6.2 | Head CT scan done within 2 hours of arrival at hospital for a non-transferred patient with Glasgow Coma Scale score <8 and systolic blood pressure >90. | 10.0 | 2 | Yes |
| | 14 | Operative treatment of gunshot wound to the abdomen. | 9.0 | 1 | No |
| | | | 10.0 | 2 | No |
| | *15.1* | *Fixation of femoral fracture in an adult patient within 24 hours of arrival to emergency department.* | *6.5* | *1* | *No* |
| | *15.2* | *Fixation of isolated closed femoral shaft fracture in an adult patient within 24 hours of arrival to emergency department.* | *7.0* | *2* | *Yes* |
| LMIC | 21.1 | Vital signs recorded within 5 min of arrival to emergency department (must include breathing assessment, heart rate, blood pressure, oxygen saturation if available). | 10.0 | 1 | Yes |
| | 21.2 | Vital signs recorded within 5 min of arrival to emergency department (must include breathing assessment, heart rate, blood pressure, oxygen saturation). | 10.0 | 2 | Yes |
| | 22 | Senior medical officer made aware of the patient with difficulty breathing, or shock (HR >100 or SBP <110) present at triage or oxygen saturation <95% within 5 min of initial assessment. | 10.0 | 1 | No |
| | | | 10.0 | 2 | No |
| | 23 | The clinician did assess airway patency by asking the patient a question and listening for a response. | 10.0 | 1 | No |
| | | | 10.0 | 2 | No |
| | 24.1 | Basic airway manoeuvre assistance (ie, sweep, chin lift, jaw thrust, oral or nasal airway, suction) performed for a patient with difficulty or obstructed breathing. | 10.0 | 1 | No |
| | 24.2 | Basic airway manoeuvre assistance (ie, jaw thrust, oral or nasal airway, suction, removal of foreign object) performed for a patient with difficulty or obstructed breathing. | 10.0 | 2 | Yes |
| | 25 | Examination for pneumothorax-haemothorax done by listening to both sides of the chest with a stethoscope within 5 min of patient arrival to emergency department. | 10.0 | 1 | Yes |
| | | | 10.0 | 2 | Yes |
| | 26 | Chest tube placed within 30 min of patient arrival in a patient with suspected or confirmed pneumothorax or haemothorax and oxygen saturation less than 98%. | 9.0 | 1 | No |
| | | | 10.0 | 2 | No |
| | 27.1 | Large-bore intravenous was placed within 5 min of patient arrival to the emergency department. | 10.0 | 1 | Yes |
| | 27.2 | Large-bore intravenous was placed within 5 min of patient arrival to the emergency department in patients with tachycardia (heart rate >110) or hypotension (systolic blood pressure <90). | 10.0 | 2 | Yes |
| | 28 | Pressure applied to external bleeding at patient arrival to the emergency department, and maintained until definitive control is performed. | 10.0 | 1 | No |
| | | | 10.0 | 2 | No |
| | 30.1 | Reduction and/or splinting with analgesia made for a long-bone fracture within 2 hours of admission or prior to transfer. | 10.0 | 1 | No |
| | 30.2 | Splinting with analgesia made for a long-bone fracture within 30 min of admission or prior to transfer. | 10.0 | 2 | Yes |
| | 35.1 | Burn patient did receive 2–4 mL of crystalloid solution per kilogram body weight per per cent body surface burn within 24 hours of injury. | 10.0 | 1 | No |
| | 35.2 | Burn patient did receive 4 mL of Ringer's lactate per kilogram body weight per per cent body surface burn within 24 hours of injury. | 10.0 | 2 | Yes |
| | 36.1 | Senior attending physician alerted when airway is compromised, using jaw thrust, chin lift, oropharyngeal/nasopharyngeal airway or suction to open airway. | 10.0 | 1 | No |
| | 36.2 | Senior attending physician alerted within 5 min of patient arrival to the emergency department when airway is compromised, usage jaw thrust, chin lift, oropharyngeal/nasopharyngeal airway or suction to open airway. | 10.0 | 2 | Yes |
| | 37 | Assessment of mouth/throat for foreign bodies and debris made in a patient who has difficulty breathing, within 10 min of arrival to emergency department. | 10.0 | 1 | No |
| | | | 10.0 | 2 | No |
| | 38.1 | Breathing assessment made within 15 min of arrival to emergency department. | 10.0 | 1 | No |
| | 38.2 | Breathing assessment made within 5 min of arrival to emergency department. | 10.0 | 2 | Yes |
| | 40.1 | Patient assessed for hypovolaemia when presenting with hypotension and tachycardia or suspected intra-abdominal bleeding, femoral shaft fracture or pelvic fracture. | 10.0 | 1 | No |

**Table 2** Continued

| Group* | ID | Audit filter | Score† | Site | Modified‡ |
|---|---|---|---|---|---|
| | 40.2 | Patient assessed for hypovolaemia using clinical examination, USG (Ultrasonography), FAST (Focused assessment with sonography in trauma) or DPL (Diagnostic peritoneal lavage) within 15 min of arrival to the emergency department when presenting with hypotension and tachycardia or suspected intra-abdominal bleeding, femoral shaft fracture or pelvic fracture. | 10.0 | 2 | Yes |
| | *45.1* | *Laparotomy done within 1 hour of arrival to the emergency department in a patient with abdominal injuries and systolic blood pressure <90.* | *5.5* | *1* | *No* |
| | 45.2 | Laparotomy done within 1 hour of arrival to the emergency department in a patient with abdominal injuries and systolic blood pressure <90 after fluid resuscitation. | 10.0 | 2 | Yes |
| | 46.1 | Immobilisation and imaging performed in a patient with suspected spine injury, within 4 hours of arrival to the emergency department. | 10.0 | 1 | No |
| | 46.2 | Immobilisation within 10 min and imaging performed within 4 hours of arrival to the emergency department in a patient with suspected spine injury. | 10.0 | 2 | Yes |
| | 47 | Intravenous antibiotics given within 1 hour of arrival to the emergency department in a patient with an open fracture. | 10.0 | 1 | No |
| | | | 10.0 | 2 | No |
| | 49.1 | Operation for irrigation and debridement within 12 hours from arrival to emergency department for an open fracture. | 10.0 | 1 | No |
| | 49.2 | Operation for irrigation and debridement within 6 hours from arrival to emergency department in a haemodynamically stable patient with an open fracture. | 10.0 | 2 | Yes |
| | 58.1 | Intubation performed in a patient with a GCS score of 8 or less within 30 min of arrival to emergency department. | 10.0 | 1 | No |
| | 58.2 | Intubation performed in a patient with a GCS score of 8 or less within 10 min of arrival to emergency department. | 10.0 | 2 | Yes |
| | *62* | *Operation for subdural or epidural haematoma within 3 hours of arrival to emergency department.* | *8.0* | *1* | *No* |
| | | | 9.0 | 2 | No |
| | 65 | FAST examination performed within 30 min from arrival to the emergency department to exclude haemoperitoneum. | 10.0 | 1 | No |
| | | | 9.0 | 2 | No |
| New | *66.1* | *Antibiotics used in acute major (50% or more) burns within 24 hours.* | *6.0* | *1* | *NA* |
| | 66.2 | Response time of respective department in attending the call. | 9.0 | 2 | NA |
| | 67.1 | MESS(Mangled extremity severity score) or WHO trauma scale used in prognosis mangled upper extremity. | 9.0 | 1 | NA |
| | *67.2* | *Sample sent for investigations.* | *8.0* | *2* | *NA* |
| | *68.1* | *Facial 3D scan done to rule out facial fractures within 24 hours of arrival to the emergency department.* | *5.0* | *1* | *NA* |
| | 68.2 | Airway breathing and circulation assessed immediately on arrival of a patient to the emergency department. | 10.0 | 2 | NA |
| | 69.1 | Serial assessment of vitals and GCS after admission. | 10.0 | 1 | NA |
| | 69.2 | Oxygen therapy with simple face mask initiated in a patient whose $SpO_2$ is less than 92% within 5 min of initial assessment of the patient. | 10.0 | 2 | NA |
| | 70 | Response time in initiating definite treatment from arrival to the emergency department, by specialist department, within 1 hour from arrival to the emergency department. | 9.0 | 2 | NA |
| | 71 | Blood components started within 4 hours of arrival to the emergency department if the patient has an Hb <70 g/L. | 9.0 | 2 | NA |
| | 72 | AVPU(Alert, verbal, pain, unresponsive) for initial assessment and followed by sequential GCS. | 9.0 | 2 | NA |
| | *73* | *Response time of surgeons.* | *8.0* | *2* | *NA* |
| | *74* | *Inotropes in a patient with shock.* | *6.0* | *2* | *NA* |
| | *75* | *Drug-assisted intubation.* | *6.5* | *2* | *NA* |
| | 76 | Sample sent for blood group and cross-match in patients with significant bleeding (heart rate >110 or systolic blood pressure <90) within 15 min of arrival to the emergency department. | 10.0 | 2 | NA |

*Original filter source group.
†Final median score.
‡Indicates whether the filter was modified by the panellists.
GCS, Glasgow Coma Scale; HR, heart rate; LMIC, low and middle-income country; NA, not applicable; SBP, systolic blood pressure.

valid for a substantial part of the patient population but specific enough to state what interventions need to be done, when they are to be done and which patients. All themes and high-level subcategories are found in box 1.

**Assessing medical relevance**

Filters perceived as medically relevant received high usefulness scores, and filters not perceived as medically relevant received lower scores. This may indicate that one of the main characteristics of an audit filter, for it to be

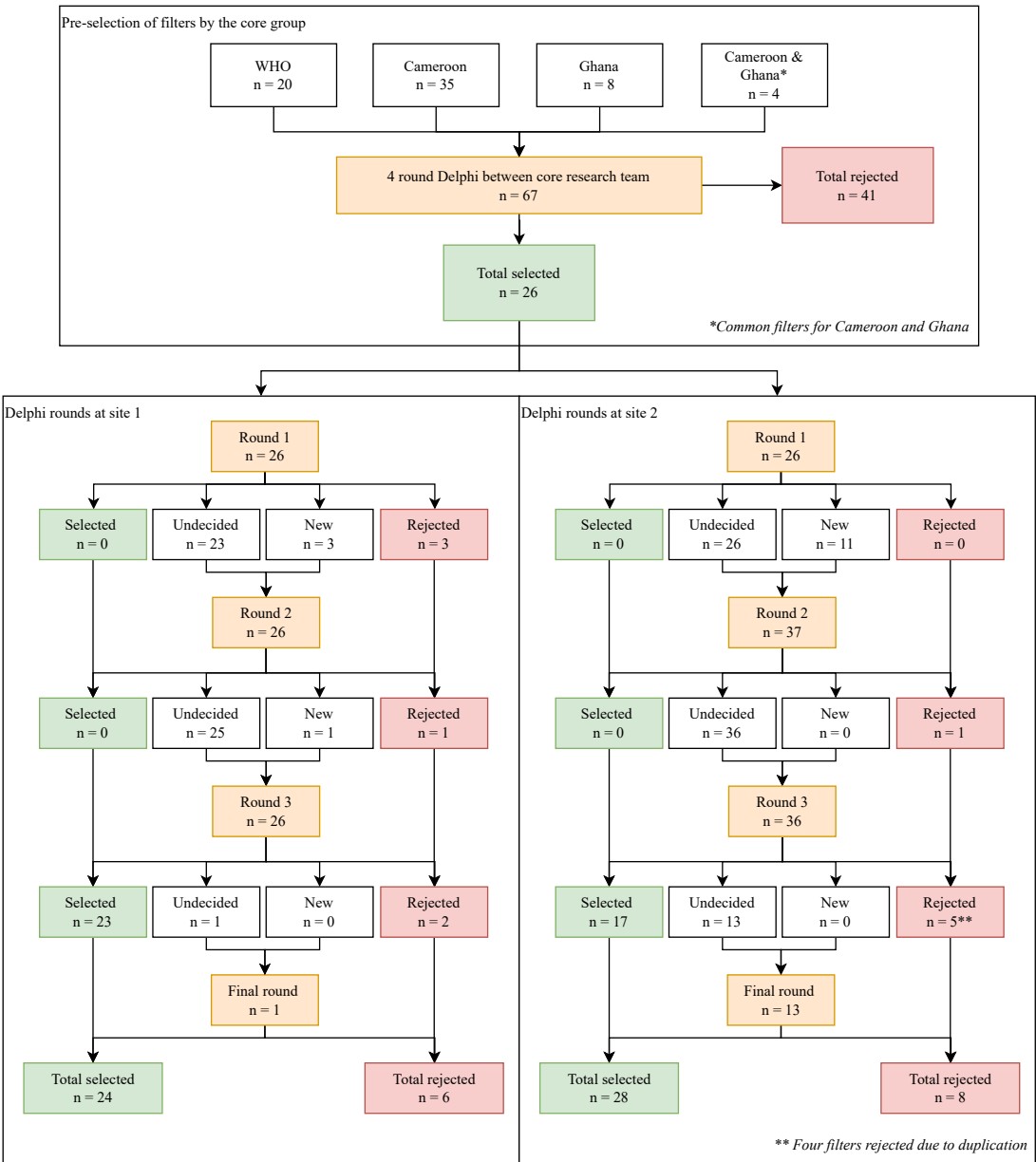

**Figure 2** All Delphi iterations and filter selection based on identified filters, by site.

perceived as useful, is that it is perceived as medically relevant. If considered relevant the filters could score highly, even if they had difficult feasibility problems. Filters that originated from WHO were to a higher extent discussed as being less medically relevant than filters from LMIC sources.

For example, discussions on filter 65: 'FAST exam performed within 30 minutes from arrival to the emergency department to exclude hemoperitoneum':

*"Subject to availability (remark: of the ultrasound machine), may be dropped at the moment…"*

## Box 1    Identified themes during content analysis of comments

**Main themes and subcategories**
**Theme 1: assessing medical relevance**
Filter deemed medically important.
Modifying filters based on a medical rationale.
Questions medical relevance or correctness.
**Theme 2: questioning feasibility**
Concern that blame will be on the provider.
Filter describes something ideal—but not achievable.
Logistic limitations.
Lacking equipment.
Lacking intensive care unit (ICU) beds.
Lacking operation theatre.
Lacking staff.
Time frame not feasible.
Modifying filters to make them more achievable.
Make the filter more conservative.
Limiting interventions only for patients who need it.
Task shifting.
Everyone on the trauma team can do simple interventions.
**Theme 3: increasing filter specificity**
Argues for higher specificity of the filter.
Defining cut-offs.
Defining target population.
Defining time to intervention.
Defining wherein the process filter should apply.
Defining who should do what.

*"FAST should stay as an audit filter as it's integral."*

*"Don't attempt to reframe this extremely useful filter. Instead channel energies to get FAST for emergency."*

In the above example, there is a difficult feasibility problem, the lack of ultrasound machines. However, there was agreement that this intervention is so medically relevant that the filter should be implemented anyway and be used to advocate for ultrasound resources in the emergency department.

### Feasibility

In some instances, participants agreed that filters described something medically ideal but not deemed feasible. Reasons for this were lack of resources such as equipment or staff or feasibility in data collection.

For example, while discussing filter 62, 'Operation for sub or epidural hematoma within 3 hours of arrival to emergency department', panellists commented:

*"Feasibility of performing emergency operation within the desired time frame of 3 hours of arrival for this condition remains an issue."*

On occasion, if the filter was deemed medically important, it was modified to make it more achievable. This could be done by restricting the patient population that should receive the intervention or prolonging the time to intervention while balancing this towards the medical relevance of the intervention. For example, filters that

describe simple airway manoeuvres should be done immediately, whereas a filter stating time to CT may be prolonged.

For example, filter 6 was originally 'Head computerized tomography (CT) scan done within 2 hours of arrival at hospital for a non-transferred patient with Glasgow Coma Scale score <13 and systolic blood pressure >90'. At one site, this was modified from a cut-off of Glasgow Coma Scale (GCS) score <13 to GCS score <8 causing a significant change in the target population, increasing the likelihood of significant pathology and reducing the intended use of CT. After this, the filter received high scores and reached a consensus.

### Filter specificity

The discussions within this theme reflected that filters could have a varying degree of specificity, that is, how precise they were in defining a target population, an intervention and often a time to that intervention. These cut-offs, to define the target population and time to intervention, were heavily discussed. The cut-offs may not be backed by a high degree of medical evidence and hence become based on consensus.

Example, filter 27: 'Large bore IV was placed within 5 min of patient arrival to the emergency department in patients with tachycardia (heart rate >110) or hypotension (systolic blood pressure <90)'.

This filter was agreed on after four iterations. It contains several cut-offs, one that defines a time to intervention (within 5 min), and two thresholds defining the target population should receive the intervention (patients with heart rate >110 or systolic blood pressure <90).

This theme intersects with the theme of making filters feasible for implementation because making a filter more conservative may limit the number of patients exposed to an intervention or allow for more time to intervention, making the filter feasible in this context. However, agreeing on specific limits and framing to increase the specificity of the filter was not always done to make it more feasible.

Filter specificity was heavily discussed for filters with high and low usefulness scores within all groups, indicating that specificity is a general characteristic of an audit filter and may not be what deems it as being useful.

### DISCUSSION

We found that the filters defined in LMIC had significantly higher usefulness scores and were deemed highly medically relevant compared with filters from the WHO. One reason for this may be that the filters suggested by WHO are derived mainly from filters in mature trauma systems in HIC.[7 8] Filters from LMIC may be reflective of trauma systems more similar to the urban India context.

The American College of Surgeons Committee on Trauma (ACSCOT) published the first audit filters for trauma care improvement in 1987. These 12 filters were derived by expert opinion, and since then, they have

evolved and been applied in several trauma systems.[8] Over time, it has become clear that filters need to be adapted to local environments, and since 2006, ACSCOT does not list specific filters but recommends filter tracking depending on local priorities. The WHO guidelines also emphasise that the filters listed are potential and that their usefulness may depend on local circumstances.[7] Agreeing on national or worldwide quality indicators for trauma care is a challenge, and the perception of what indicators are deemed important varies between regions.[19]

The audit filters defined in Cameroon[9] and Ghana[10] were deemed highly useful in the urban Indian setting. The filters from Cameroon were focused on regional referral hospitals, while those defined in Ghana were intended for use in district-level hospitals. Many of these filters represent the initial management, resuscitation and simple, potentially life-saving interventions like airway manoeuvres or placement of intravenous catheters. These filters may not be perceived as useful in a mature trauma system where these interventions are a part of routine care, firmly built in by years of improving training, processes and access to equipment.

Several studies from HICs have struggled to determine the effect on outcomes of specific audit filters. The review process itself, being costly and labour intensive, has raised questions about the usefulness of audit filters in well-developed trauma systems.[8 20] Their potential use in less developed trauma systems is less studied but seems promising. One study by Chadbunchachai et al in Thailand showed a significant reduction of preventable trauma deaths between 1994 and 1997 after implementing trauma audit filters.[5 21] The discrepancy that the same filter may be useful and potentially influence outcomes in one context and not the other highlights the importance of audit filters being selected and evolved locally to suit the context of their use.

Hypothetically, applying filters developed in HICs and mature trauma systems to a low-resource setting may be disruptive since they may divert focus from the more pressing concerns and hence not improve the processes that could affect outcomes in that context.

The LMIC definition is broad; we have chosen to use it here as a distinction against HIC with mature trauma systems. Cameroon, Ghana and India would be classified more specifically as lower middle-income countries. For low-income countries, there is very limited research on trauma systems.[4] Whether the filters evaluated in this study would be transferable to low-income countries is unclear. Treatment of a majority of trauma patients requires a healthcare infrastructure that can supply surgical services. This is lacking in several contexts, especially in low-income and lower middle-income countries and in rural areas.[22]

We found that the new suggested filters at both sites had the lowest median scores. This may reflect the difficulty of defining audit filters based on consensus with online survey interaction only. Developing new audit filters that are clear, feasible and specific is a complex task. If this is attempted via an online survey, it may be beneficial to supply the participants with a short suggestion on how an audit filter may be formulated to make sure it is specific, highlights the medical relevance and is feasible in the context.

One of this study's main strengths is the nesting within the TAFT project since both sites performed the surveys to select audit filters for implementation in the hospital. The use of two separate sites adds strength to our quantitative analysis, as both sites showed similar results. The online Delphi approach enabled us to execute a study at two sites in different cities, with a wide range of participants at a very low cost. Similar studies could be replicated in other low-resource settings with standard technical devices.

There are several limitations to our study. First, the preselection of audit filters made by the core team might have induced bias and contributed to the overall high usefulness scores of the filters. However, analysis of the results of this preselection Delphi did not reveal any significant difference in usefulness scores between filters from WHO and LMIC during the preselection (online supplemental figure 1). Second, the number of filters from the WHO was lower, and one filter receiving low scores had a large impact on the mean for this group. Third, the statistical analysis is based on a convenience sample. The inclusion of participants does not represent a randomised sample, and the overall high scores contribute to a ceiling effect which may affect the results of the statistical analysis. Fourth, the generalisability of the selected filters is limited because of the local modifications to the filters, based on a smaller number of experts at each site. Lastly, the content analysis was done based on comments, not by directed questions. It was not mandatory to leave a comment, and hence this discussion primarily focuses on problems or features to improve. Because the main focus of the discussion was to agree on something standardised and measurable, very little about experiences or deeper mechanisms can be extracted from this material. We believe, however, that the content analysis gives another level of understanding as to why filters received the scores that they did, which may help guide efforts when adapting audit filters in other contexts.

We also want to highlight that our expert panels mainly consisted of surgeons and physicians, and perspectives from other healthcare personnel were limited. Management of the trauma patient involves a broad team of personnel and including panellists who reflect this is important for future studies in this area.

## CONCLUSION

The trauma audit filters defined in Ghana and Cameroon were deemed highly useful in the urban India context, indicating that the potential for transferability of trauma audit filters between similar contexts is high. These filters may be used as a starting ground when implementing trauma improvement programmes including audit filters in similar settings. Compared with filters derived from

high-resource programmes, many of these filters focus on initial interventions and resuscitation and were deemed to have high medical relevance in this context.

**Author affiliations**
[1]Department of Global Public Health, Karolinska Institutet, Stockholm, Sweden
[2]Department of Emergency and Internal Medicine, Skane University Hospital, Malmo, Sweden
[3]The George Institute for Global Health India, New Delhi, Delhi, India
[4]Emergency Medicine, Department of Clinical Sciences, Lund University Faculty of Medicine, Lund, Sweden
[5]Department of Surgery, Maulana Azad Medical College, New Delhi, Delhi, India
[6]WHO Collaboration Centre for Research in Surgical Care Delivery In Low and Middle-Income Countries, Mumbai, Maharashtra, India
[7]Department of Surgery, Seth Sukhlal Karnani Memorial Hospital, Kolkata, West Bengal, India
[8]Department of Surgery, RG Kar Medical College, Kolkata, West Bengal, India
[9]Doctors For You, Mumbai, Maharashtra, India
[10]Department of Radiodiagnosis and Imaging, Maulana Azad Medical College, New Delhi, Delhi, India
[11]School of Public health, Tata Institute of Social Sciences, Mumbai, Maharashtra, India
[12]Department of Surgery, Seth GS Medical College and KEM Hospital, Mumbai, Maharashtra, India
[13]Department of ENT and Head & Neck Surgery, Maulana Azad Medical College, New Delhi, Delhi, India
[14]Surgery, Vydehi Institute of Medical Sciences and Research Centre, Bangalore, Karnataka, India
[15]School Public Health and Community Medicine, Institute of Medicine, University of Gothenburg Sahlgrenska Academy, Goteborg, Sweden
[16]Department of Accident and Emergency, Lok Nayak Hospital, New Delhi, India
[17]Department of Burns and Plastic Surgery, Maulana Azad Medical College, New Delhi, Delhi, India
[18]Department of Critical and Intensive Care, All India Institute of Medical Sciences, New Delhi, Delhi, India
[19]Department of Orthopaedic Surgery, Maulana Azad Medical College, New Delhi, Delhi, India
[20]Function Perioperative Medicine and Intensive Care, Karolinska University Hospital, Stockholm, Sweden

**Twitter** Nobhojit Roy @#nobsroy

**Acknowledgements** We would like to thank all the panellists in the Delphi survey for their time and dedication to this project. Their constructive work in refining the audit filters is what made this study possible. A warm thank you to the participating centres, where the efforts by a great many continue as these filters are implemented and continuously used for review. Finally, thank you to all the participants in the ongoing Trauma Audit Filter Trial: clinicians, healthcare staff, project officers and mostly all patients and relatives who generously allow us to include them.

**Contributors** MGW, NR, KDS, MK conceived the study as part of the Trauma Audit Filter Trial. JB, MGW, HMA, NR, KDS, MK contributed to the design of the study. MGW, NR, KDS, MK and SD conducted the on-site training before the online Delphi. JB, MGW, NR, LB, PKB, SC, SD, SG, MK, PL, VM, RM, AM, LNM, RS, PS, RS, KDS and SS were a part of the execution of the online Delphi. JB prepared the data. JB and MGW conducted the statistical analysis. MP reviewed the statistical analysis. JB performed the content analysis in collaboration with HMA and MGW. JB and MGW drafted the first version of the article. The interpretation of data and critical revisions of the work for important intellectual content was performed by JB, MGW, HMA, NR, LB, PKB, SC, SD, SG, MK, PL, VM, RM, AM, LNM, RS, PS, RS, KDS, SS, MP, UE and JK. MGW is the guarantor.

**Funding** This work was supported by the Swedish Research Council (2016-02041).

**Competing interests** None declared.

**Patient and public involvement** Patients and/or the public were not involved in the design, or conduct, or reporting, or dissemination plans of this research.

**Patient consent for publication** Not required.

**Ethics approval** Ethical approvals exist as a part of the TAFT project: Swedish Ethical Review Authority, 2017/930-31/2, dated 7 June 2017; MAMC—F.1/IEC/MAMC/(57/02/2017/No113), dated 19 July 2017; SSKM—IPGME&R/IEC/2017/396, dated 21 August 2017.

**Provenance and peer review** Not commissioned; externally peer reviewed.

**Data availability statement** Data are available in a public, open access repository. Data and R code for the quantitative analysis are publicly available on GitHub at https://github.com/titco/utaf. To protect the privacy of the participants the qualitative data are not available.

**ORCID iDs**
Johanna Berg http://orcid.org/0000-0001-7553-7337
Helle Molsted Alvesson http://orcid.org/0000-0001-6109-7203
Nobhojit Roy http://orcid.org/0000-0003-2022-7416
Lovenish Bains http://orcid.org/0000-0002-8627-0452
Lakshmeswar Nagaraj Mohan http://orcid.org/0000-0002-3503-832X
Martin Gerdin Wärnberg http://orcid.org/0000-0001-6069-4794

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
