## [Reviewer comments · BMJ Open]

ARTICLE DETAILS

TITLE (PROVISIONAL)	Perceived usefulness of trauma audit filters in urban India: a mixed-methods multicenter Delphi study comparing filters from the World Health Organization and Low-and-Middle-Income Countries
AUTHORS	Berg, Johanna; Alvesson, Helle; Roy, Nobhojit; Ekelund, Ulf; Bains, Lovenish; Chatterjee, Shamita; Bhattacharjee, Prosanta; Siddarth, David; Gupta, Swati; Kamble, Jyoti; Khajanchi, Monty; Lal, Pawanindra; Malhotra, Vikas; Meher, Ravi; Mishra, Anurag; Mohan, L N; Petzold, Max; Saxena, Ritu; Shrivastava, Prabhat; Singh, Rajdeep; Soni, Kapil; Sural, Sumit; Gerdin Wörnberg, Martin

VERSION 1 – REVIEW

REVIEWER	Heino, Anssi Turku University Hospital, Anaesthesia and Intensive Care
REVIEW RETURNED	14-Jan-2022

GENERAL COMMENTS	Comments to the Author Reviewer: Anssi Heino, MD, PhD Department of Perioperative Services, Intensive Care Medicine and Pain Management, Turku University Hospital, Turku, Finland Comments to the Author, Thank you for this opportunity to comment on the manuscript. As authors describe, trauma audit filters are widely recommended to be used in trauma services to secure and follow the quality of care. At the same time, a lack of evidence exists on the benefits of these filters. This issue is well described and discussed by the authors. Moreover, the focus being on the usability of these trauma audit filters in low- and middle-income countries. The authors raise an important fact: trauma audit filters should be specifically implemented for the service they are used in, and a filter used in a high-income country service, will not necessarily suite a service with a lack of resources –for a number of reasons, all of which are well described in the manuscript. Please, see the following specific sectional comments. Specific comments: ABSTRACT Well build. Summarizes the manuscript. INTRODUCTION Informative and compact, well written.
--

	MATERIAL AND METHODS This section is overall well written. However, couple of questions do raise regarding the study methods and statistics. These being rather crucial regarding the whole manuscript. Identification of audit filters. The selection of trauma audit filters –LMIC vs. WHO vs. NEW- is justified, and well presented in the manuscript. As is the use of TAFT core team and their separate Delphi rounds. However, why did the authors choose LMIC filters from Cameroon and Ghana? What was the reasoning behind this? There are wide range of LMICs` that could have been used as audit filter sources for the study, so perhaps, consider adding few short explanations why these two countries were used. Was it because similar healthcare, economical background, etc. compared to India? Participants The expert selections for these type of studies is challenging, and selection bias can never be fully overcome. It is described in the manuscript, but authors could add a bit more details of the inclusions criteria, and especially if any participants were excluded, or exclusion criteria used? A total number of 27 individuals (25 actual participants) is a fair number for this type of study, but how did the authors decide this number? Was any kind of study strength analysis used? Moreover, when the authors studied the participants as two separate groups (site 1 and site 2), the actual number of participants for the analysis were 10 and 15. As these were analyzed as individual study groups, the number of experts on a Delphi round is markedly less. The online surveys, consensus and stability criteria The use of online surveys is well described and widely used method in these type of studies. Likert-Scale as well, is an ingenious choice. However, some major questions are raised regarding this section. First, why the two sites were studied and analyzed separately? Why not as a one, and this way markedly more powerful panel of experts. As the existing literature describes, one limitation in Delphi studies is how highly opinionial individuals can overcome other less opinionial individuals in an expert groups. This is a greater risk in small expert groups, for instance, in this study setting: a group of 10 experts (site 1) on an expert group, opposite to a possible 25 (site 1 + 2) experts group. The two groups could have been put on one Delphi, instead of two separate, and by this handle the biases and reach a stronger expert panel. Second, why Wilcoxon-rank-sum test was used in a Delphi setting? This could be reasonable, but needs a bit more explanation –not necessarily in the manuscript. Why not, for instance, an agreement percent and intraclass correlation coefficient to reach an expert consensus on Delphi rounds? Statistical analysis As mentioned earlier, the decision to use Wilcoxon-rank-sum test and Kruskal-Wallis to reach an expert consensus has to be explained. Again, not necessarily in the manuscript but to reason it for the Editor. Content analysis Well described, and justified. RESULTS
--	---

	Results are well described, as per used methods. However, methodology and statistical analysis has to be reasoned a bit more, as described on earlier comments. Table two is highly informative in this sections, but the tables are commented more detailed later. Yet, as the two sites were studied separately, there are now clinically confusing results seen. For example table 2, where ID 3.1 and 3.2 are close to each other, but whether GCS evaluation is done every 30 minutes or hourly is a major clinical difference. Or 30.1 and 30.2, with 30 minutes vs. 2 hours to splint a long bone fracture. These type of results actually limit the clinical usability of the study findings in trauma systems rather dramatically. Content analysis Well described. The sections: Assessing medical relevance, feasibility and filter specificity are extremely informative, and describe the issues regarding audit filters in an excellent form by using examples. DISCUSSION Well written and summarizes the findings of this current study. Even more importantly, set the study findings in a context regarding the overall use of audit filters. Especially the discussion on how the audit filters could be more beneficial in less developed trauma systems, is important, and raises the importance for future studies. CONCLUSION Well written. TABLES, FIGURES AND ADDITIONAL MATERIAL Tables 1., 2. and 3. are highly informative but should operate in individual units, so that a reader could get the information without reading the whole manuscript. Therefore, textual explanations should be added on all of these. For instance, in the footnotes and headings of the tables. Figure 1. is extremely important for a reader to understand the study setting. But as the figures, needs a bit more information in the footnotes/heading. Figure 2. Relevant and informative in its existing form. Supplemental material is informative and relevant in its existing form.
--	--

REVIEWER	Berube, Melanie CHU de Quebec-Universite Laval
REVIEW RETURNED	25-Jan-2022

GENERAL COMMENTS	Abstract:  -Please spell out the abbreviations. -Not clear what is meant by audit filters...were deemed highly useful in this context. What context? Introduction:  -The filter Patient with a GCS < 8 is intubated within 2 hours is not an optimal example since this is not a recommended practice. -There is inconsistency in what is said about the evidence in relation to the audit filters. On the one hand, it is mentioned in line 11 that there is evidence that implementing trauma quality improvement programs decreases mortality and improves trauma care in both HIC
--

and LMIC. On the other hand, it is mentioned in line 20 that High-quality evidence on the effects on outcomes after implementing trauma audit filters is lacking.

-It is not clear why the authors brought up that audit filters require more extensive resources... Those resources are often not available in LMIC. Therefore, mentioning the resources aspect diminishes the relevance of the suggested study.

-Please identify the sources of audit filters at the end of the introduction.

-Was the third objective determine a priori or was it added according to what emerged from participants' comments? I do not consider the analysis of comments as an objective since no question was specifically address on characteristics of audit filters that may impact their usefulness. I would rather present related findings as characteristics to take into account in the implementation or in the evaluation of site performance.

Methods:

-Design: It is mentioned in the abstract that a mixed-methods study was used, but not in the methods. Please specify what was the quantitative data and the qualitative data and how they relate to the objectives.

-Explain why a Delphi was used for the preselection of audit filters among a small group of experts. Was it because there were too many audits to evaluate?

-The response rates from participants and table 1 should be presented in the results section. Considering the small number of participants by categories, I would rather present the characteristics of the sample in the text and not in a Table.

-A Likert scale from 1-10 is uncommon. Please provide the questionnaire in supplemental digital file.

-The evaluation of stability with the Wilcoxon-rank-sum test is explained in 2 sections. It should be mentioned in the statistical analysis section only.

-Analyses were performed by site. This should be stated in the methods. Moreover, the authors should explain what led them to use this approach. Why not combining results from both sites to select indicators applicable to all urban centers in India? This is usually the approach we used in trauma systems.

-Content analysis: the section starting in line 53 to line 7 on the subsequent page is not clear. What is the defined groups? Please review since it is difficult to understand what was really done as qualitative analysis.

Results:

-The order of findings presentation should be revised to follow the objectives 1 to 3: Perceived usefulness, generate-context specific audit filters...

-Specify what is "all three groups".

-It is not clear why the perceived usefulness of LIMC, WHO and NEW filters was compared if at the end filters from each of these groups were selected. I am not convinced that these analyses provide relevant information.

-The section on content analysis needs to be revised. It is very difficult to keep up with the flow of ideas. Also, I would focus on audit filters that have been selected to explain the characteristics that must be considered while implementing them and not on filters that were not selected.

Discussion:

Lines 38 to 47: The argument that audit filters do not appear to be effective in HICs but could be effective in LMICs is contradictory. Why would these audit filters work in LMICs and not in HICs that have multiple resources and that have worked for 40 years on implementing quality indicators in organized trauma systems? In addition, I suggest reviewing the literature of Dr. Lynne Moore who has demonstrated the benefits of using audit filters in HICs. I suggest adding this reference as well when providing background to trauma quality indicators:

https://www.researchgate.net/publication/350064100_Trauma_quality_indicators_internationally_approved_core_factors_for_trauma_management_quality_evaluation_the_WSES_Trauma_Quality_Indicators_Expert_Panel

	-It is mentioned that having conducted the Delphi in 2 sites was a study strength. How exactly is this a strength considering that each site has been studied individually and the scores are not combined to get a better idea of what to implement in India? Having targeted only 2 sites could also be seen as a limitation in terms of generalizing the results. -I suggest to better delineate the limitations. The authors start with firstly...but what were the second, third...limits. Also, limits need to be named then explained. Conclusion: -The medical relevance was brought up in the last part of the conclusion. What about feasibility and specificity?
--	---

VERSION 1 – AUTHOR RESPONSE

Reviewer: 1 Dr. Anssi Heino, Turku University Hospital Comments to the Author: Comments to the Author

Comments to the Author,

Thank you for this opportunity to comment on the manuscript. As authors describe, trauma audit filters are widely recommended to be used in trauma services to secure and follow the quality of care. At the same time, a lack of evidence exists on the benefits of these filters. This issue is well described and discussed by the authors. Moreover, the focus being on the usability of these trauma audit filters in low- and middle-income countries. The authors raise an important fact: trauma audit filters should be specifically implemented for the service they are used in, and a filter used in a high-income country service, will not necessarily suite a service with a lack of resources –for a number of reasons, all of which are well described in the manuscript. Please, see the following specific sectional comments.

Thank you for taking the time to read our manuscript and provide valuable feedback!

Specific comments:

ABSTRACT Well build. Summarizes the manuscript.

Thank you!

INTRODUCTION Informative and compact, well written.

Thank you!

MATERIAL AND METHODS This section is overall well written. However, couple of questions do raise regarding the study methods and statistics. These being rather crucial regarding the whole manuscript.

Identification of audit filters. The selection of trauma audit filters –LMIC vs. WHO vs. NEW- is justified, and well presented in the manuscript. As is the use of TAFT core team and their separate Delphi rounds. However, why did the authors choose LMIC filters from Cameroon and Ghana? What was the reasoning behind this? There are wide range of LMICs` that could have been used as audit filter sources for the study, so perhaps, consider adding few short explanations why these two countries were used. Was it because similar healthcare, economical background, etc. compared to India?

Thank you for pointing this out to us to help us better clarify this. At the time of this study, these two papers were the only ones known to the authors to have been published in peer-

reviewed journals. It may well be that audit filters may have been used in other, similar contexts but not been published. The inclusion of these articles was based on a small literature review that did not include grey literature. We have made a minor update in the manuscript under Methods -> Identification of audit filters from:

“We identified trauma audit filters from Cameroon[9], Ghana[10] and those proposed in the WHO guidelines.[7] These were selected based on their intended use in LMIC.” **to**

“We included previously published trauma audit filters intended for use in LMICs. These filters included those from the WHO guidelines[7], as well as filters in one study from Cameroon[9] and one from Ghana[10].”

Participants The expert selections for these type of studies is challenging, and selection bias can never be fully overcome. It is described in the manuscript, but authors could add a bit more details of the inclusions criteria, and especially if any participants were excluded, or exclusion criteria used? A total number of 27 individuals (25 actual participants) is a fair number for this type of study, but how did the authors decide this number? Was any kind of study strength analysis used? Moreover, when the authors studied the participants as two separate groups (site 1 and site 2), the actual number of participants for the analysis were 10 and 15. As these were analyzed as individual study groups, the number of experts on a Delphi round is markedly less.

Thank you for highlighting this important point! There are two major reasons for the participation size. Firstly as the objective was narrow, to rate and select filters for implementation at specific hospitals, we wanted each group to be represented by participants who knew the possibilities and barriers of implementing these filters at the local hospital specifically. Even at tertiary hospitals, the number of people with this experience is limited. Secondly, since we allowed for modification and addition of filters, we felt that having a too-large group would be feasible in the online format. Based on this we made work experience within the area of trauma care our only inclusion criteria and no exclusion criteria were used. We estimated that a number between 10 and 15 participants at each hospital would be feasible. We did not conduct any power calculation a priori as the main purpose of significance testing during the Delphi was “informal”, to determine the stability of ratings between rounds.

We have elaborated on this in the Methods section -> Participants, from:

“We chose this approach since considerable knowledge about the local conditions and organisation, in addition to work experience, would be necessary to evaluate the usefulness of the filters at each site.” **to**

“We chose this approach since considerable knowledge about the local conditions and organisation, in addition to work experience, would be necessary to evaluate the usefulness of the filters at each site. Because we also allowed the filters to be modified and new filters to be suggested, we deemed 10-15 participants feasible to reach a consensus. “

The online surveys, consensus and stability criteria The use of online surveys is well described and widely used method in these type of studies. Likert-Scale as well, is an ingenious choice. However, some major questions are raised regarding this section. First, why the two sites were studied and analyzed separately? Why not as a one, and this way markedly more powerful panel of experts. As the existing literature describes, one limitation in Delphi studies is how highly opinionated individuals can overcome other less opinionated individuals in an expert groups. This is a greater risk in small expert groups, for instance, in this study setting: a group of 10 experts (site 1) on an expert group, opposite

to a possible 25 (site 1 + 2) experts group. The two groups could have been put on one Delphi, instead of two separate, and by this handle the biases and reach a stronger expert panel.

Thank you for this excellent question to help us elaborate on these important points. We agree that a wider group of experts would reduce the risk of the consensus being driven by one or two opinionated individuals and that the risk of this increases with smaller groups. We had two major reasons to conduct these Delphis at different sites:

- 1. Our first objective was to compare the usefulness scores for different filters based on their source/origin. We perceive it as a strength to be able to have two independent groups of experts at two hospitals rate these filters to allow us to see if there were similar results in the usefulness scores at both sites independent of each other.***
- 2. The second objective was to choose and create appropriate filters for implementation at specific hospitals. Even though both hospitals are large tertiary hospitals in urban India, local challenges and possibilities vary between the hospitals, which may require specific modifications to the filters. We wanted each hospital to select filters based on their local needs.***

We do agree that if the objective of our study had been to supply a list of audit filters that could be used as a general recommendation in India, consensus from a larger diverse group would have been a requirement. We have added the following to the Methods -> Design section as a fourth paragraph:

We performed two independent Delphi surveys at two different hospitals to allow for the audit filters to be selected and modified based on local priorities and capabilities at each hospital. This design also allowed us to compare scores between two independent groups of experts and see if results were reproducible across the two study sites.

Second, why Wilcoxon-rank-sum test was used in a Delphi setting? This could be reasonable, but needs a bit more explanation –not necessarily in the manuscript. Why not, for instance, an agreement percent and intraclass correlation coefficient to reach an expert consensus on Delphi rounds?

Thank you for highlighting this. Defining consensus criteria for Delphi studies is still, to our knowledge, a challenging area where there are limited guidelines and a lot of heterogeneity in the published literature. The a priori definition of consensus was based on the need to be able to select filters with very high scores (Median of 9 and above) and reject filters with low scores. Stability was determined for each filter and round using the non-parametric Wilcoxon-test, as the data from Likert ratings were not normally distributed nor were the number of ratings for each filter enough to allow for parametric testing. The Wilcoxon was chosen due to its extensive use in these conditions, even though not being the most common in Delphi studies. We believe ICC could also have been used to determine stability in a similar fashion.

Statistical analysis As mentioned earlier, the decision to use Wilcoxon-rank-sum test and Kruskal-Wallis to reach an expert consensus has to be explained. Again, not necessarily in the manuscript but to reason it for the Editor.

Thank you for pointing this out! The Wilcoxon-rank-sum test was used for determining stability for each filter during rounds. The Kruskal-Wallis was used in the analysis of the results to allow us to compare the scores over three different groups using a nonparametric test. Since this requires a post hoc test, we followed this with the wilcoxon-test, as it's suitable for comparing groups of non-normal distributed data. We chose the Bonferroni adjustment for multiple tests, being a conservative correction method limiting the risk of us overestimating the results. Since we use Wilcoxon both to determine stability for each filter and round and our

final result analysis, we have tried to clarify this in the manuscript by renaming Methods -> Statistical analysis to Methods - Statistical analysis of the results as well as a minor change in the text in the second paragraph from:

We used the Kruskal-Wallis test to compare usefulness scores across the three groups. If this resulted in a significant difference ($p < 0.05$), further analysis was conducted using the Wilcoxon-rank-sum test to find differences between the individual groups using a significance level of $p < 0.05$. **To**

We used the Kruskal-Wallis test to compare usefulness scores across the three groups. If this resulted in a significant difference ($p < 0.05$), post hoc testing was conducted using the Wilcoxon-rank-sum test to identify differences between the individual groups using a significance level of $p < 0.05$.

Content analysis Well described, and justified.

Thank you!

RESULTS Results are well described, as per used methods. However, methodology and statistical analysis has to be reasoned a bit more, as described on earlier comments. Table two is highly informative in this sections, but the tables are commented more detailed later. Yet, as the two sites were studied separately, there are now clinically confusing results seen. For example table 2, where ID 3.1 and 3.2 are close to each other, but whether GCS evaluation is done every 30 minutes or hourly is a major clinical difference. Or 30.1 and 30.2, with 30 minutes vs. 2 hours to splint a long bone fracture. These type of results actually limit the clinical usability of the study findings in trauma systems rather dramatically.

Thank you for raising this important point! We agree, we believe that these differences reflect the difficulty to create audit filters with these levels of details. It also highlights the different local priorities and challenges and what is being deemed feasible. Optimally, all these filters would be based on high-quality medical evidence but a lot of the cutoffs are based on clinical experience and expert consensus. Creating these filters with a local focus limits their generalizability. We feel this is very important to highlight and have added this in the 9th paragraph of the discussion that lists limitations:

“Fourthly, the generalisability of the selected filters is limited because of the local modifications to the filters, based on a smaller number of experts at each site.”

Content analysis Well described. The sections: Assessing medical relevance, feasibility and filter specificity are extremely informative, and describe the issues regarding audit filters in an excellent form by using examples.

Thank you!

DISCUSSION Well written and summarizes the findings of this current study. Even more importantly, set the study findings in a context regarding the overall use of audit filters. Especially the discussion on how the audit filters could be more beneficial in less developed trauma systems, is important, and raises the importance for future studies.

Thank you!

CONCLUSION Well written.

TABLES, FIGURES AND ADDITIONAL MATERIAL Tables 1., 2. and 3. are highly informative but should operate as individual units, so that a reader could get the information without reading the whole manuscript. Therefore, textual explanations should be added on all of these. For instance, in the footnotes and headings of the tables. Figure 1. is extremely important for a reader to understand the study setting. But as the figures, needs a bit more information in the footnotes/heading. Figure 2. Relevant and informative in its existing form. Supplemental material is informative and relevant in its existing form.

Thank you for pointing this out. We have updated the headings and supplied footnotes for the tables as well as additional clarification in figure 1.

Reviewer: 2 Prof. Melanie Berube, CHU de Quebec-Universite Laval, Universite Laval Comments to the Author: Abstract:

-Please spell out the abbreviations.

Thank you, we agree and as suggested we have spelt out the abbreviations where they appear first time in the abstract.

-Not clear what is meant by audit filters...were deemed highly useful in this context. What context?

Thank you for pointing this out, we have now clarified this sentence by defining the context as urban India in the abstract from:

Conclusions: Audit filters from other LMIC were deemed highly useful in this context. **To**

Conclusions: Audit filters from other LMIC were deemed highly useful in the urban India context.

Introduction:

-The filter Patient with a GCS < 8 is intubated within 2 hours is not an optimal example since this is not a recommended practice.

Thank you for reflecting on this! We agree that the time frame for use in this filter was poorly chosen, and we have updated it to reflect the way the filter was originally framed in the WHO guidelines in the second paragraph of the Introduction from:

"For example, an audit filter could state, "Patient with a GCS < 8 is intubated within 2 hours". " **To**

"For example, an audit filter could state, "Patient with Glasgow Coma Scale score <8 should have an endotracheal tube or surgical airway performed before leaving resuscitation area."

-There is inconsistency in what is said about the evidence in relation to the audit filters. On the one hand, it is mentioned in line 11 that there is evidence that implementing trauma quality improvement programs decreases mortality and improves trauma care in both HIC and LMIC. On the other hand, it is mentioned in line 20 that High-quality evidence on the effects on outcomes after implementing trauma audit filters is lacking.

Thank you for highlighting this. To the best of our knowledge, there is evidence to support that the implementation of trauma quality improvement programs (TQIP) reduces mortality and improves trauma care in both these settings. However, these programs may involve various approaches to trauma quality improvement with other interventions like the implementation of trauma registries or systems. When it comes to the specific evidence on trauma audit filters, there's no high-quality evidence of this specific intervention and its effect on outcomes. We have attempted to clarify this in the second paragraph of the introduction from:

High-quality evidence on the effects on outcomes after implementing trauma audit filters is lacking.[6] **to**

Even though there is evidence that TQIP improves outcomes, high-quality evidence on the effects of trauma audit filters, is lacking.[6]

-It is not clear why the authors brought up that audit filters require more extensive resources... Those resources are often not available in LMIC. Therefore, mentioning the resources aspect diminishes the relevance of the suggested study.

Thank you for raising this important point. In the WHO guidelines, it's reflected that the use of audit filters requires more resources than some of the other proposed interventions, since tracking audit filters requires an infrastructure for data collection and data-driven review meetings. We feel that it is important to highlight this especially when we are proposing interventions to be used in LMICs. Transparency on this is essential to help decision-makers make informed choices about interventions, where most of them have limited evidence in this setting.

-Please identify the sources of audit filters at the end of the introduction.

Thank you, we agree that this will improve the reader's understanding early on and have updated the aim to directly convey the sources of audit filters in both the abstract and the introduction. The first aim was modified from:

With this background, this study aims to compare experts' perceived usefulness of audit filters from different sources **to**

With this background, this study aims to compare experts' perceived usefulness of audit filters from Ghana, Cameroon, WHO and those locally developed

-Was the third objective determine a priori or was it added according to what emerged from participants' comments? I do not consider the analysis of comments as an objective since no question was specifically address on characteristics of audit filters that may impact their usefulness. I would rather present related findings as characteristics to take into account in the implementation or in the evaluation of site performance.

Yes, the objective was determined a priori. We wanted to assess whether we could identify characteristics of these audit filters based on what was discussed during the Delphi by the participants. For example, a lot of discussions revolved around agreeing on cut-offs, and we identified this as a characteristic that is a part of the theme's specificity. Secondly, we wanted to compare whether these characteristics correlated to the usefulness scores that the filters had received. Hence, characteristics here are common traits that we identified in this process. We felt that this was best captured by allowing the conversation to score, select and modify filters take place without any specific guidance or targeted questions.

Methods:

-Design: It is mentioned in the abstract that a mixed-methods study was used, but not in the methods. Please specify what was the quantitative data and the qualitative data and how they relate to the objectives.

Thank you for clarifying this. We have added this to Methods - Design in paragraph 2 from:

"We used the Delphi technique to facilitate consensus within a panel of experts." **To**

"We conducted a mixed-methods study based on a Delphi technique to facilitate consensus within a panel of experts."

and we added the third paragraph:

“This mixed-methods design allowed us to gather quantitative data from the scoring of the audit filters to answer the question of whether perceived usefulness differed based on the original source as well as select the highest scoring filters for implementation. The qualitative data, the written comments deliberated during the Delphi rounds, was used to explore characteristics of the audit filters and their correlation to the usefulness scores.”

-Explain why a Delphi was used for the preselection of audit filters among a small group of experts. Was it because there were too many audits to evaluate?

That is correct. The purpose of the preselection was

- 1) To remove duplicate filters.***
- 2) Prioritise filters, as 67 filters were not deemed feasible to score and comment at each site.***
- 3) Evaluate the technical tool used for Delphi.***

We have attempted to clarify this under Methods -> Identification of audit filters from:

“The TAFT core research team, based in India, conducted an internal online Delphi survey to remove duplicate filters, prioritize filters deemed potentially appropriate for the urban setting in India and evaluate the online tool used for this trial.” **To**

“We did not deem it feasible to score and comment on all 67 filters at each site and therefore the India based TAFT core research team conducted an internal online Delphi survey to remove duplicate filters, prioritise filters potentially appropriate for the urban setting in India and evaluate the online tool used for this study.”

-The response rates from participants and table 1 should be presented in the results section.

Thank you, we have moved the third paragraph from Methods -> Participants to Results -> Participants along with table 1.

- Considering the small number of participants by categories, I would rather present the characteristics of the sample in the text and not in a Table.

Thank you for this suggestion. We did try to summarise this as text but felt that it became more difficult to get an overview of the sample and compare the sites. We, therefore, decided to keep the table.

-A Likert scale from 1-10 is uncommon. Please provide the questionnaire in supplemental digital file.

Thank you, we have added the questionnaire used in the first round of the Delphi as supplementary material.

-The evaluation of stability with the Wilcoxon-rank-sum test is explained in 2 sections. It should be mentioned in the statistical analysis section only.

Thank you for pointing this out. We realised that this was not clearly described in the manuscript. The Wilcoxon-rank-sum test was used both to determine stability, explained in the section describing consensus and stability. It was also used in the quantitative statistical analysis when comparing the scores between the different groups, outlined in the statistical analysis section. We have attempted to clarify this in the manuscript under Methods by renaming the section “Statistical analysis” to “Statistical analysis of the results” and a minor modification in the second paragraph of this section from:

“We used the Kruskal-Wallis test to compare usefulness scores across the three groups. If this resulted in a significant difference ($p < 0.05$), further analysis was conducted using the Wilcoxon-rank-sum test to find differences between the individual groups using a significance level of $p < 0.05$.” **To**

“We used the Kruskal-Wallis test to compare usefulness scores across the three groups. If this resulted in a significant difference ($p < 0.05$), post hoc testing was conducted using the Wilcoxon-rank-sum test to identify differences between the individual groups using a significance level of $p < 0.05$.”

-Analyses were performed by site. This should be stated in the methods. Moreover, the authors should explain what led them to use this approach. Why not combining results from both sites to select indicators applicable to all urban centers in India? This is usually the approach we used in trauma systems.

Thank you for pointing this out. There are two main reasons why we chose to do this as two independent surveys.

1) Our first objective was to compare whether the origin of the audit filter impacted the perceived usefulness in this context. We wanted to see if two independent groups at two different hospitals in this setting would end up with similar results.

2) Our second objective was to select audit filters for implementation at two hospitals. Even if both of these are large tertiary hospitals, they have local challenges and processes in place that may influence what filters were chosen and how they needed to be locally adapted.

We agree that if our main objective was to create a general consensus-based list of audit filters to be used as a baseline in urban India, then a larger joint multicenter Delphi would have been a requirement. We believe that this was not well described and have added the following to the Methods -> Design section as a fourth paragraph:

“We performed two independent Delphi surveys at two different hospitals to allow for the audit filters to be selected and modified based on local priorities and capabilities at each hospital. This design also allowed us to compare scores between two independent groups of experts and see if results were reproducible across the two study sites.”

-Content analysis: the section starting in line 53 to line 7 on the subsequent page is not clear. What is the defined groups? Please review since it is difficult to understand what was really done as qualitative analysis.

Thank you. We have attempted to expand and clarify this section in greater detail as well as describing the groups (high/low scores, and source groups previously defined, WHO, LMIC and New) and the steps in the analysis. We have updated section Methods -> Content analysis paragraph one from:

“We explored the characteristics of the audit filters that were discussed using inductive content analysis.[16]”

To

“We explored the characteristics of the audit filters that were discussed using inductive content analysis.[16] We chose this approach as there are limited studies that have described characteristics of audit filters. The team of authors represent a multi-disciplinary group of clinical, social science and epidemiological expertise. Several are physicians and surgeons at participating hospitals.”

And Methods -> Content analysis paragraph two, from:

“All codes were sorted into categories that were abstracted into themes. After this, we assessed how the themes and categories compared to usefulness scores and the defined groups. We did this by selecting a theme or a category, looking at which group of filters these were derived from to see if one group was more frequently represented than another. If this was the case, we went back to the comments associated with this theme and group to understand the specific discussion better. We used this frequency analysis to help find patterns in the comments that would have been difficult to detect.”

to

“All codes were sorted into categories that were abstracted into themes. After this, we assessed how the themes and categories compared to usefulness scores and the three source groups; WHO, LMIC and New. We did this by defining groups of low-scoring and high-scoring filters, based on their final filter scores. We then selected a theme or high-level category and assessed which filters and filter groups the theme was associated with. We used frequency analysis to see if the theme may be dominant in any of the specified groups. If this was the case, we returned to the comments associated with this theme and group to understand the specific discussion, from this perspective, better. We used this frequency analysis to help find patterns in the comments that would have been difficult to detect. We developed an audit trail as the coding was discussed between authors JB, HMA and MGW during the process of analysis. “

Results:

-The order of findings presentation should be revised to follow the objectives 1 to 3: Perceived usefulness, generate-context specific audit filters...

Thank you for this suggestion. We have revised the manuscript by changing the headings of the sections in the Result section from:

Results -> Perceived usefulness, content analysis **to**

Results -> Perceived usefulness, selection of audit filters, characteristics of audit filters in relation to their usefulness

-Specify what is “all three groups”.

Thank you. This has been clarified in the manuscript under Results -> Perceived usefulness from:

“Comparing all three groups, we found significant differences in usefulness scores at both sites.” **To**

"Comparing all three source groups; WHO, LMIC and New, we found significant differences in usefulness scores at both sites "_

-It is not clear why the perceived usefulness of LIMC, WHO and NEW filters was compared if at the end filters from each of these groups were selected. I am not convinced that these analyses provide relevant information.

Thank you for this reflection. Our first objective was to compare if audit filters from different origins/sources differed in terms of how useful they were perceived to be in this setting. To do this, we combined filters into the three groups (LMIC, WHO and New) and compared the scoring among these. Filters may well be selected from any of the groups since the filters in themselves are also heterogeneous and represent different parts of the in-hospital trauma care process. However, our objective was to find out if their perceived usefulness overall differed between these groups.

-The section on content analysis needs to be revised. It is very difficult to keep up with the flow of ideas. Also, I would focus on audit filters that have been selected to explain the characteristics that must be considered while implementing them and not on filters that were not selected.

Thank you. Our objective was to explore what characteristics were associated with usefulness scores, both high and low. We did find some correlation, for example, that filters that were considered highly medically relevant were associated with high scoring filters. If we were only to include filters that were selected, we would not be able to find traits associated with lower scores or with no association to scores.

Discussion:

Lines 38 to 47: The argument that audit filters do not appear to be effective in HICs but could be effective in LMICs is contradictory. Why would these audit filters work in LMICs and not in HICs that have multiple resources and that have worked for 40 years on implementing quality indicators in organized trauma systems?

Thank you for raising this excellent point. Our hypothesis is that the use of audit filters may have a larger effect on clinical outcomes in a less developed trauma system than in a more developed one - because the issues and deviations from care, leading to patient mortality and morbidity, may be vastly different between these contexts. Many issues have already been identified and attended to in a developed system. In a context where no trauma system is in place, these filters may prove valuable to shift attention and resources to identified problems. Exactly how these mechanisms work, what processes that are actually improved by implementing audit filters in this setting, is largely unknown. But in a health care system with no trauma registry and no ongoing data collection, implementing audit filters may reveal issues that might be considered basic in a mature system. The audit filters from Ghana and Cameroon reveal a focus on initial resuscitation and basic interventions like airway manoeuvres and placement of IVs. These may be things that are not tracked in a mature trauma system simply because they are a part of standard care, firmly built-in by years of improving training and processes. We have attempted to clarify this in the third paragraph of the discussion, from:

"The filters from Cameroon were focused on regional referral hospitals, while those defined in Ghana were intended for use in district-level hospitals. Many of these filters represent the initial management, resuscitation and simple, potentially life-saving interventions. These filters may not be perceived as useful in a mature trauma system where these interventions are assumed."

To

“The filters from Cameroon were focused on regional referral hospitals, while those defined in Ghana were intended for use in district-level hospitals. Many of these filters represent the initial management, resuscitation and simple, potentially life-saving interventions like airway manoeuvres or placement of intravenous catheters. These filters may not be perceived as useful in a mature trauma system where these interventions are a part of routine care, firmly built-in by years of improving training, processes and access to equipment.”

In addition, I suggest reviewing the literature of Dr. Lynne Moore who has demonstrated the benefits of using audit filters in HICs. I suggest adding this reference as well when providing background to trauma quality indicators:

<https://www.researchgate.net/publication/350064100_Trauma_quality_indicators_internationally_approved_core_factors_for_trauma_management_quality_evaluation_the_WSES_Trauma_Quality_Indicators_Expert_Panel>

Thank you for highlighting the important work by Dr. Lynne Moore and providing this excellent reference. We have added this reference to the second paragraph of the Discussion on the background of trauma audit filters, from:

“Over time it has become clear that filters need to be adapted to local environments, and since 2006, ACSCOT does not list specific filters but recommends filter tracking depending on local priorities. The WHO guidelines also emphasize that the filters listed are potential and that their usefulness may depend on local circumstances.[7]”

To

“Over time it has become clear that filters need to be adapted to local environments, and since 2006, ACSCOT does not list specific filters but recommends filter tracking depending on local priorities. The WHO guidelines also emphasise that the filters listed are potential and that their usefulness may depend on local circumstances.[7] Agreeing on national or worldwide quality indicators for trauma care is a challenge, and the perception of what indicators are deemed important varies between regions.[18]”

-It is mentioned that having conducted the Delphi in 2 sites was a study strength. How exactly is this a strength considering that each site has been studied individually and the scores are not combined to get a better idea of what to implement in India? Having targeted only 2 sites could also be seen as a limitation in terms of generalizing the results.

Thank you for highlighting this. We consider this a strength since our first objective was to study whether there was a difference in usefulness scores between audit filters from different sources. By having two independent groups of experts replicating similar results, we consider this a strength in reference to this objective. We agree that if the main objective would have been to generate audit filters to be a baseline for implementation in India, this would have been an unsuitable approach. Creating generalisable audit filters for India would require a different study design with multiple centres and regions taking part in one Delphi to find consensus. We have made a minor addition to the 8th paragraph of the manuscript from:

“The use of two separate sites adds strength to our results.” **To**

The use of two separate sites adds strength to our quantitative analysis, as both sites showed similar results.

-I suggest to better delineate the limitations. The authors start with firstly...but what were the second, third...limits. Also, limits need to be named then explained.

Thank you, we fully agree and have revised the manuscript accordingly by modifying paragraph 10 and merging it with paragraph 12 in the Discussion, from:

“There are several limitations to our study. Firstly, the pre-selection of audit filters made by the core team may induce a bias and may have contributed to the overall high usefulness scores of the filters. However, analysis of the results of this pre-selection Delphi did not reveal any significant difference in usefulness scores between filters from WHO and LMIC during the pre-selection. (Supplemental material: Figure 1) The actual number of filters from the WHO was also lower, and one filter receiving low scores had a large impact overall. The statistical analysis is based on a convenience sample, not a truly randomized sample, but we believe that the significance testing is still of value. The overall high scores contribute to a ceiling effect which may affect our results.” **To**

“There are several limitations to our study. Firstly, the pre-selection of audit filters made by the core team might have induced bias and contributed to the overall high usefulness scores of the filters. However, analysis of the results of this pre-selection Delphi did not reveal any significant difference in usefulness scores between filters from WHO and LMIC during the pre-selection. (Supplemental material: Figure 1) Secondly, the number of filters from the WHO was lower, and one filter receiving low scores had a large impact on the mean for this group. Thirdly, the statistical analysis is based on a convenience sample. The inclusion of participants do not represent a randomized sample, and the overall high scores contribute to a ceiling effect which may affect the results of the statistical analysis. Fourthly, the generalisability of the selected filters is limited because of the local modifications to the filters, based on a smaller number of experts at each site.”

Conclusion:

-The medical relevance was brought up in the last part of the conclusion. What about feasibility and specificity?

Thank you for pointing this out. Our results show that the filters that originated from other LMICs were perceived as highly useful. The content analysis also revealed that these filters were believed to have high medical relevance in this setting. We believe that this may be the reason for the high scoring of these filters, our conclusion is that this points to a transferability between contexts with similar challenges. The themes feasibility and specificity describe interesting characteristics of audit filters and their adoption process but do not seem to correlate to-usefulness scores therefore we have not included them in our conclusion.

Reviewer: 1 Competing interests of Reviewer: No competing interests.

Reviewer: 2 Competing interests of Reviewer: Potentially intellectual

VERSION 2 – REVIEW

REVIEWER	Heino, Anssi Turku University Hospital, Anaesthesia and Intensive Care
REVIEW RETURNED	08-Apr-2022

GENERAL COMMENTS	The authors have revised the manuscript, and well addressed all the questions raised by the reviewers.
--

REVIEWER	Berube, Melanie CHU de Quebec-Universite Laval
REVIEW RETURNED	18-Apr-2022

GENERAL COMMENTS	The authors have addressed the previous suggestions well overall. I suggest numbering the study objectives when presenting them for clarity.
--